EMBO
Molecular Medicine

# Spironolactone is an antagonist of NRG1-ERBB4 signaling and schizophrenia-relevant endophenotypes in mice

Michael C Wehr[1,†,*] (ID), Wilko Hinrichs[2,†], Magdalena M Brzózka[1], Tilmann Unterbarnscheidt[2,3], Alexander Herholt[4], Jan P Wintgens[4], Sergi Papiol[1,5], Maria Clara Soto-Bernardini[2], Mykola Kravchenko[6], Mingyue Zhang[6], Klaus-Armin Nave[2], Sven P Wichert[1], Peter Falkai[1], Weiqi Zhang[6], Markus H Schwab[2,3,7] & Moritz J Rossner[1,2,**] (ID)

## Abstract

Enhanced NRG1-ERBB4 signaling is a risk pathway in schizophrenia, and corresponding mouse models display several endophenotypes of the disease. Nonetheless, pathway-directed treatment strategies with clinically applicable compounds have not been identified. Here, we applied a cell-based assay using the split TEV technology to screen a library of clinically applicable compounds to identify modulators of NRG1-ERBB4 signaling for repurposing. We recovered spironolactone, known as antagonist of corticosteroids, as an inhibitor of the ERBB4 receptor and tested it in pharmacological and biochemical assays to assess secondary compound actions. Transgenic mice overexpressing Nrg1 type III display cortical Erbb4 hyperphosphorylation, a condition observed in postmortem brains from schizophrenia patients. Spironolactone treatment reverted hyperphosphorylation of activated Erbb4 in these mice. In behavioral tests, spironolactone treatment of Nrg1 type III transgenic mice ameliorated schizophrenia-relevant behavioral endophenotypes, such as reduced sensorimotor gating, hyperactivity, and impaired working memory. Moreover, spironolactone increases spontaneous inhibitory postsynaptic currents in cortical slices supporting an ERBB4-mediated mode-of-action. Our findings suggest that spironolactone, a clinically safe drug, provides an opportunity for new treatment options for schizophrenia.

**Keywords** drug repositioning; NRG1-ERBB4; schizophrenia; spironolactone; split TEV assay
**Subject Categories** Neuroscience; Pharmacology & Drug Discovery

## Introduction

Schizophrenia (SZ) is as severely debilitating neuropsychiatric disorder characterized by positive symptoms, that is, hallucinations and delusions, negative symptoms, that is, lack of motivation, and cognitive symptoms (Insel, 2010). Positive symptoms can frequently be ameliorated by treatment with dopamine receptor antagonists, but efficient treatment options for negative and cognitive symptoms are not available (Goff et al, 2011). Thus, there is a strong clinical need to develop and explore more target-directed therapies for SZ (Nestler & Hyman, 2010). Repurposing of existing drugs principally offers a fast track to the clinic and has been demanded for SZ (Insel, 2012; Lencz & Malhotra, 2015), also because many pharma companies withdrew from research on severe mental disorders (Margraf & Schneider, 2016). Genetic association studies have identified NRG1 and its cognate receptor ERBB4 as SZ risk genes, and altered NRG-ERBB4 signaling has been associated with positive, negative, and cognitive symptoms (Stefansson et al, 2002; Harrison & Law, 2006; Li et al, 2006; Nicodemus et al, 2006). Several postmortem studies revealed increased expression of NRG1 in SZ patients (Hashimoto et al, 2004; Law et al, 2006; Weickert et al, 2012). Elevated expression of the ERBB4-JM-a-CYT-1 variant carrying a PI3K-recruitment domain has also been detected in SZ (Silberberg et al, 2006; Law et al, 2007, 2012). Moreover, ERBB4 was found to be

1   Molecular and Behavioral Neurobiology, Department of Psychiatry, Ludwig Maximilian University of Munich, Munich, Germany
2   Department of Neurogenetics, Max Planck Institute of Experimental Medicine, Göttingen, Germany
3   Cellular Neurophysiology, Center of Physiology, Hannover Medical School, Hannover, Germany
4   Systasy Bioscience GmbH, Munich, Germany
5   Institute of Psychiatric Phenomics and Genomics (IPPG), Medical Center of the University of Munich, Munich, Germany
6   Laboratory of Molecular Psychiatry, Department of Psychiatry, University of Münster, Münster, Germany
7   Center for Systems Neuroscience (ZSN), Hanover, Germany
    *Corresponding author. Tel: +49 89 4400 53275; Fax: +49 89 4400 55853; E-mail: michael.wehr@med.uni-muenchen.de
    **Corresponding author. Tel: +49 89 4400 55891; Fax: +49 89 4400 55853; E-mail: moritz.rossner@med.uni-muenchen.de
    †These authors contributed equally to this work

   

hyperphosphorylated in postmortem brains from SZ patients (Hahn *et al*, 2006), suggesting that NRG1-ERBB4 hyperstimulation might represent a component of SZ pathophysiology.

In agreement, transgenic mice with increased *Nrg1* expression display SZ-relevant behavioral deficits, including hyperactivity, impaired sensorimotor gating, decreased social interaction, and reduced cognitive functions (Deakin *et al*, 2009, 2012; Kato *et al*, 2010). In particular, transgenic mice with neuronal overexpression of the membrane-bound cysteine-rich-domain (CRD) type III isoform of Nrg1 (*Nrg1*-tg) display chronic ErbB4 hyperphosphorylation in the cortex, which is associated with a broad spectrum of SZ-relevant endophenotypes, including dysbalanced excitatory and inhibitory neurotransmission, altered spine growth, and impaired sensorimotor gating (Agarwal *et al*, 2014). Moreover, it has been shown recently that endophenotypes associated with elevated Nrg1 expression are reversible in adult animals, which strongly supports the assumption that the NRG1/ERBB4 signaling system provides a valid target for pharmacological interventions (Yin *et al*, 2013; Luo *et al*, 2014). It thus appears plausible that compounds, which can re-balance the activity of the NRG1-ERBB4 signaling pathway, could represent candidates for the therapeutic treatment of schizophrenia beyond positive symptoms.

In this study, we have first developed a co-culture assay system compatible with high-throughput-screening (HTS) utilizing the split TEV technology (Wehr *et al*, 2006, 2008). We then used this assay to screen a library of clinically approved drugs in a repurposing approach to uncover new potential target specificities (Wang & Zhang, 2013), which resulted in the identification and validation of spironolactone as an inhibitor of ERBB4. Finally, we can show that spironolactone decreases phosphorylation levels of ERBB4 *in vitro* and *in vivo* and leads to an altered balance of excitation/inhibition of cortical projection neurons. Chronic spironolactone treatment ameliorates hyperactivity and reverses sensorimotor gating and working memory deficits in *Nrg1*-tg mice. Thus, spironolactone alleviates novel aspects of SZ-relevant symptoms in this mouse model of increased NRG1-ERBB4 signaling.

# Results

### A split TEV-based co-culture assay to screen for modulators of NRG1-ERBB4 signaling

Screening for modulators of NRG1-ERBB4 signaling in cell culture requires an adequate setup reflecting endogenous signaling mechanisms. According to a current model, NRG1 ligands reside in presynaptic terminals of principle pyramidal neurons, whereas ERBB4 receptors are mainly expressed at the postsynaptic density of dendrites in inhibitory interneurons (Rico & Marín, 2011). Based on this ligand-receptor configuration, NRG1 mediates juxtacrine and paracrine signaling to ERBB4. We established a cellular HTS-compatible co-culture assay, in which NRG1 was expressed in the signal-sending cell population A, whereas ERBB4 was expressed in the signal-receiving cell population B (Fig 1A). Initially, we used the full-length NRG1 type I β1a isoform that undergoes proteolytic cleavage resulting in the release of the extracellular domain, which contains the biologically active EGF-like domain (EGFld), into the extracellular space (Hu *et al*, 2006; Willem *et al*, 2006). Therefore, NRG1 type I β1a can elicit juxtacrine (non-cleaved form) and paracrine (cleaved

form) stimuli. To screen for approved small compounds that could modulate NRG1-ERBB4 signaling, we combined this co-culture assay with the split TEV protein–protein interaction technique to monitor ERBB4 activation through induced PI3K adaptor recruitment by the human ERBB4-JMa-Cyt1 variant (Fig 1A).

The functionality and robustness of the co-culture assay was investigated by co-plating increasing numbers of PC12 cells carrying a stably integrated mouse *Nrg1 type I β1a* expression cassette (Nrg1 cells, Fig EV1A for stable Nrg1 expression) with ERBB4-PIK3R1-expressing PC12 cells (split TEV assay cells). Co-culture conditions were verified using two PC12 cell populations expressing either EYFP or ECFP (Fig EV1B). A dose–response analysis showed that the assay reached a plateau of activation when 10,000 Nrg1-expressing cells were co-plated with 40,000 split TEV assay cells, with half-maximal activation at 5,000 cells (Fig 1B). Calculation of the Z' factor, a measure of HTS applicability and quality (Zhang *et al*, 1999), resulted in a value of 0.5 indicating a large separation band at screening conditions. Importantly, addition of the soluble EGFld resulted in a twofold increase of ERBB4 activation compared with Nrg1 cells alone, implying that ERBB4 activation in the co-culture assay can be decreased and increased by potential NRG1-ERBB4 inhibitors and activators, respectively (Fig 1C). In addition, dose–response assays using ERBB4-PIK3R1 split TEV assay cells only and soluble EGFld as stimulus (single-culture assay) showed stable and reproducible dose–responses that also qualified for HTS, with Z' factors between 0.56 and 0.68 for three independent assays (Fig EV1C). The specificity of the NRG1-ERBB4 co-culture assay was validated in dose–response assays using established ERBB4 inhibitors, such as lapatinib ($IC_{50}$ value at 2.61 μM, co-culture assay, Fig 1D; 0.45 μM, single-culture assay, Fig EV1D) and CI-1033 ($IC_{50}$ value at 0.01 μM, co-culture assay, Fig EV1E; 0.004 μM, single-culture assay, Fig EV1F). Taken together, these data indicate that the split TEV-based co-culture assay provides a robust platform to screen for modulators of NRG1-ERBB4 signaling.

### Screening the NIH clinical compound collection recovers spironolactone as ERBB4 receptor antagonist

We used the split TEV-based NRG1-ERBB4-PIK3R1 co-culture assay to screen two sets of the NIH Clinical Collection (NIH-NCC) containing 727 FDA-approved drugs in total (Fig 2A). From this screen, we selected a primary hit list of candidates that were at least three standard deviations away from the mean (Fig 2B for NIH-NCC set 1; Fig EV2A for NIH-NCC set 2; Dataset EV1). These candidates were then subjected to individual re-screening to eliminate off-target effects, such as toxicity and interference with assay tools, and 18 substances met these criteria and were selected for the final hit list (Appendix Fig S1 for a flowchart of all screening and validation steps; see Appendix Table S1 for final hit list). Spironolactone, a mineralocorticoid receptor (MR) antagonist formerly used as a diuretic and to treat high blood pressure (Gaddam *et al*, 2010), was recovered as top antagonist candidate (Fig 2B). In a dose–response co-culture assay using Nrg1 type I β1a, spironolactone displayed an $IC_{50}$ value of 1.0 μM, and marginal toxic effects at higher concentrations, as indicated by reduced *Renilla* luciferase readings (Fig 2C). ERBB4-specific effects were confirmed by dose–response control assays, which showed absence of spironolactone effects on assay components (Fig EV2B and C).

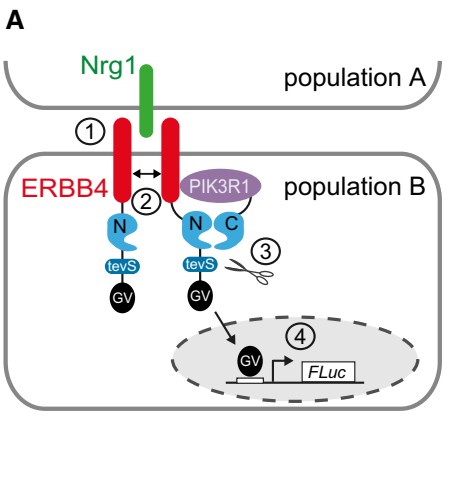

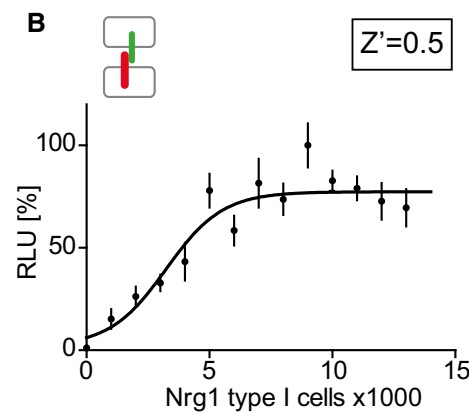

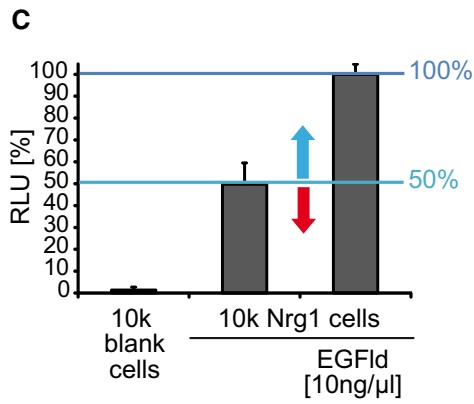

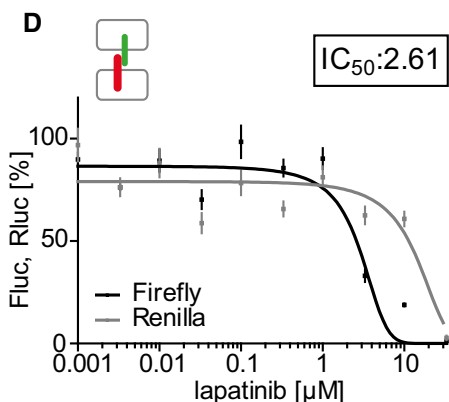

**Figure 1.  A co-culture assay based on the split TEV technique for monitoring NRG1-ERBB4 signaling activity.**

A   The ERBB4-PIK3R1 split TEV assay monitors NRG1-ERBB4 signaling in PC12 cells. The Nrg1 ligand (green) is stably expressed in the signal-sending cell population A. The signal-receiving cell population B (or split TEV assay cells) is transfected with plasmids encoding the assay components ERBB4 (red) fused to NTEV-tevS-GV (ERBB4-NTEV-tevS-GV), the adapter molecule PIK3R1 (purple, the regulatory subunit alpha of the PI3K) fused to CTEV (PIK3R1-CTEV), and a UAS-driven firefly luciferase reporter (Fluc). Upon Nrg1 binding to the extracellular domain of ERBB4 (1), ERBB4-NTEV-tevS-GV dimerizes and cross-phosphorylates itself (2). PIK3R1-CTEV binds to the phosphorylated ERBB4 receptor leading to the functional reconstitution of TEV protease activity and the concomitant release of the artificial co-transcriptional activator Gal4-VP16 (GV) through proteolytic cleavage at a TEV protease cleavage site (tevS) (3). In turn, released GV translocates to the nucleus and binds to UAS sequences (open box) to activate the transcription of a firefly reporter gene (4).

B   Dose–response assay using increasing numbers of Nrg1 type I β1a-expressing PC12 cells. For each 96-well, 40,000 split TEV assay cells were co-plated with increasing numbers of Nrg1-expressing cells and incubated for 24 h. Half-maximal activation is reached at 5,000 Nrg1-expressing cells. The Z′ factor is 0.5 indicating a large separation band for this assay.

C   Adding 10 ng/ml EGFld resulted in a twofold activation. Per 96-well, 40,000 split TEV assay cells were co-plated with empty PC12 cells (no Nrg1 expression), 10,000 Nrg1-expressing cells, and 10,000 Nrg1-expressing cells plus 10 ng/ml EGFld. Arrows indicate measuring window of activation (blue arrow) and inhibition (red arrow) relative to baseline activity.

D   Lapatinib antagonizes ERBB4-PIK3R1 signaling in a dose-dependent manner. Per 96-well, 40,000 split TEV assay cells were incubated with increasing amounts of lapatinib, followed by co-plating 10,000 Nrg1 type I β1a-expressing cells. The inset depicts the $IC_{50}$ value in μM.

Data information: After compound/stimulus addition, each assay was incubated for 24 h. RLU, relative luciferase units; Fluc, firefly luciferase activity (black line); Rluc, *Renilla* luciferase activity (gray line); $n = 6$; data are shown as mean, and error bars represent SEM.

---

Importantly, spironolactone also inhibited ERBB4 activity ($IC_{50}$ value of 1.1 μM) in a co-culture assay using the membrane-bound CRD containing type III isoform of Nrg1, the major NRG1 isoform in the brain, which is implicated in juxtacrine signaling (Fig 2D). Thus, spironolactone modulates ERBB4 activity downstream from both paracrine and juxtacrine NRG1 signaling. Further, spironolactone acts at a proximal step of ERBB4 receptor activation upstream of tyrosine phosphorylation and adapter recruitment as ERBB4 dimerization stimulated by EGFld was efficiently inhibited ($IC_{50}$ value of 1.1 μM) in a single-culture assay (Fig 2E). Finally, we used an ERBB4 variant that lacks the

intracellular domain, and thus is signaling incompetent, but expresses at the cell cortex and dimerizes upon EGFld stimulation (Fig EV3A and B). Notably, spironolactone inhibits dimerization of full-length ERBB4, but not of C-terminally truncated ERBB4 (Fig EV3C).

### Spironolactone but not its metabolic products antagonizes the ERBB4/PIK3R1 assay

Spironolactone also inhibited ERBB4 signaling activity in an ERBB4-PIK3R1 single-culture assay, albeit with a slightly increased $IC_{50}$

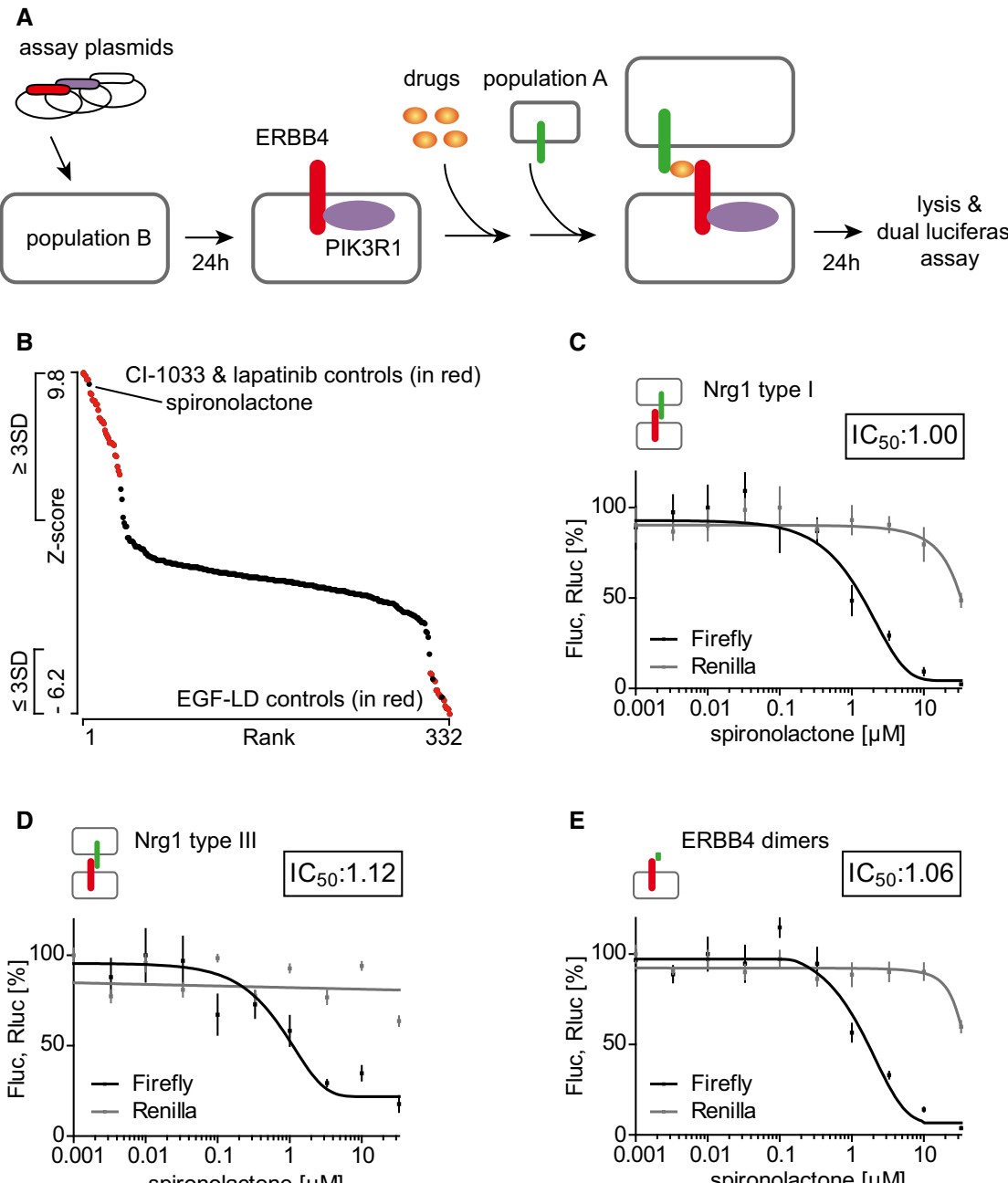

**Figure 2.  Spironolactone is the primary candidate recovered from the co-culture screen.**

A  Flow chart of the compound screen. PC12 cells (population A) were transfected in solution with the split TEV assay plasmids ERBB4-NTEV-tevS-GV and PIK3R1-CTEV and incubated for 2 h before seeded onto 96-well plates. Population A cells were allowed to express the plasmids for 24 h. Compounds were added in a concentration of 10 μM, followed by seeding the Nrg1-expressing PC12 cells (population B) half an hour later. After 24 h of compound incubation, cells were lysed and subjected to a dual luciferase assay. The screening data were analyzed using the cellHTS2 package in R Bioconductor.

B  Graphic visualization of the primary screen data of the NIH-NCC library set 1. All counts (320 compounds and 64 controls) from the Nrg1-ERBB4-PIK3R1 split TEV compound screen were plotted against the *Z*-score using the Mondrian program, with pathway activators displaying high values and inhibitors low values. For the secondary analysis, we selected all candidates that were at least three standard deviations away from the mean. EGFld-positive and lapatinib/CI-1033-negative controls are shown in red.

C, D  Spironolactone antagonizes Nrg1-ERBB4-PIK3R1 signaling. In dose–response assays using ERBB4-NTEV-tevS-GV and PIK3R1-CTEV plasmids transfected into PC12 cells, spironolactone was administered at increasing concentrations before seeding (C) Nrg1 type I- or (D) Nrg1 type III-expressing PC12 cells.

E  Spironolactone inhibits ERBB4 receptor dimerization. Dose-dependent dimerization of the ERBB4 receptor was analyzed using a split TEV assay encompassing ERBB4-NTEV-tevS-GV and ERBB4-CTEV plasmids transfected into PC12 cells. 10 ng/ml EGFld was applied as Nrg1 stimulus.

Data information: Fluc, firefly luciferase activity (black lines); Rluc, *Renilla* luciferase activity (gray lines, indicating toxicity levels); *n* = 6; data are shown as mean, and error bars represent SEM. The insets depict IC$_{50}$ values in μM.

value of 2.8 µM (Fig 3A). We used the assay to identify molecular structures in spironolactone (Fig 3A) required for the inhibition of ERBB receptor activation in this assay and examined structurally highly related compounds, such as the metabolites canrenone

(Fig 3B) and 7α-thiomethyl-spironolactone (Fig 3C) as well as the second-generation MR antagonist eplerenone (Fig 3D). Canrenone lacks the thio-ketone group attached to the sterol core structure, whereas 7α-thiomethyl-spironolactone lacks the ketone group only.

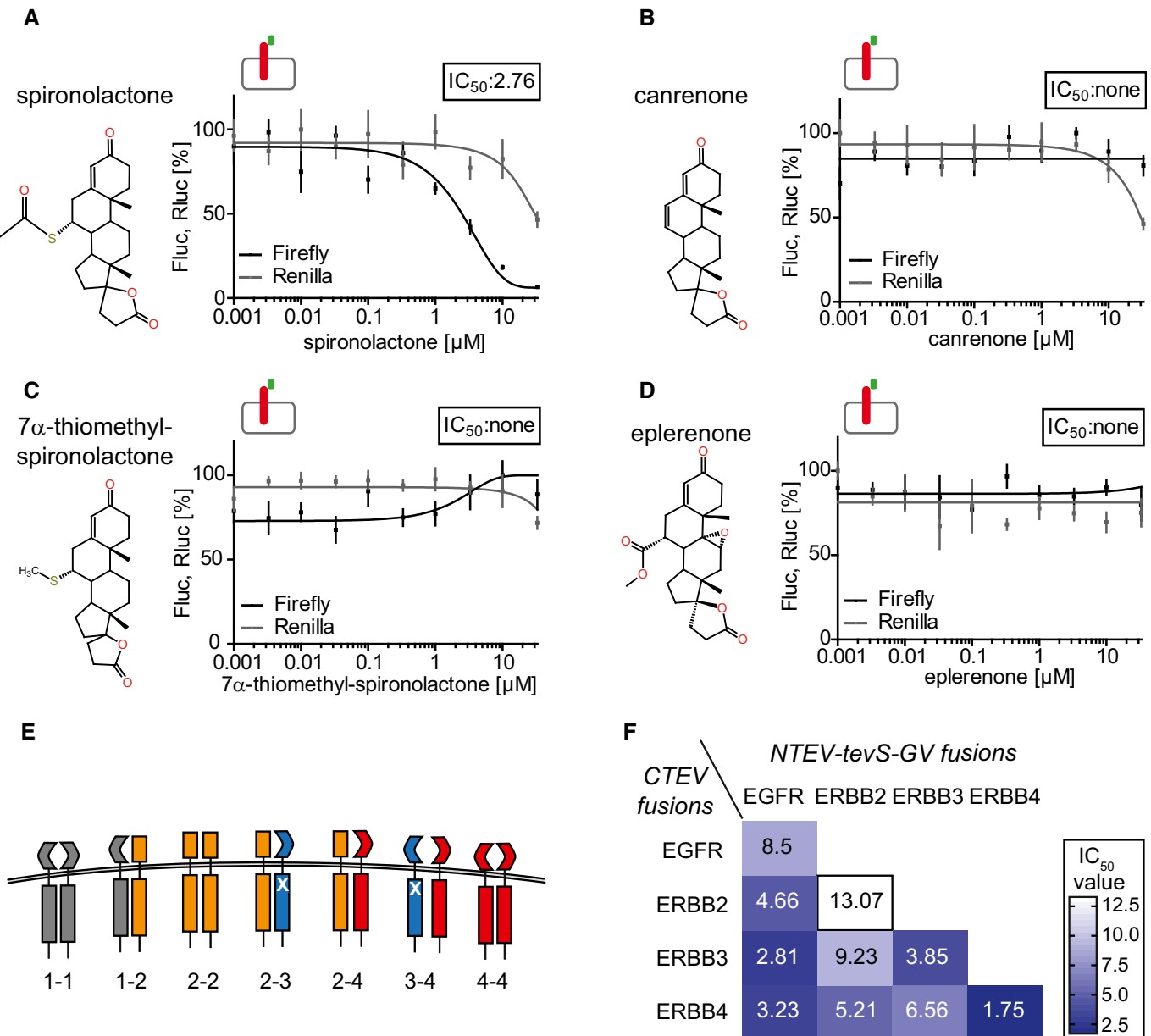

**Figure 3. Multilevel profiling approach of spironolactone treatment assessing target specificities and adapter recruitment.**

A    Spironolactone, molecular structure shown on the left, inhibits ERBB4/PIK3R1 split TEV assay activity (black line), with an IC$_{50}$ value of 2.76 µM.

B–D  The spironolactone metabolites (B) canrenone and (C) 7α-thiomethyl-spironolactone as well as the second-generation drug (D) eplerenone do not attenuate the ERBB4/PIK3R1 assay activity (black lines). Note that only spironolactone bears a thio-ketone group attached to the sterol core structure. All ERBB4/PIK3R1 assays were run in a single-culture assay mode using 10 ng/ml EGFld as functional Nrg1 stimulus, and ERBB4-NTEV-tevS-GV and PIK3R1-CTEV plasmids were transfected into PC12 cells (indicated by icon).

E    Schematic representation of the most critical ERBB dimers tested in spironolactone dose–response assays using the split TEV protein–protein interaction detection technique. Note that not all possible combinations are depicted.

F    Heat map showing the IC$_{50}$ values obtained from individual ERBB dimerization split TEV assays. All single dose–response assays can be found in Fig EV4; the combination ERBB4-ERBB4 is shown in Fig 2E.

Data information: Fluc, firefly luciferase activity (black lines, reporting ERBB4-PIK3R1 assay activity); Rluc, *Renilla* luciferase activity (gray lines, assessing viability); $n = 6$; data are shown as mean, and error bars represent SEM. The insets depict IC$_{50}$ values in µM.

In eplerenone, the thio-ketone group is replaced by an acidic group, in addition to a minor modification at the sterol core structure. None of these compounds showed an inhibitory effect in the ERBB4-PIK3R1 single-culture assay (Fig 3B–D), suggesting that the thio-ketone group specific for spironolactone plays a role for its inhibitory function in this assay.

## Spironolactone preferentially inhibits dimers containing ERBB4

Target specificity is a crucial aspect for the evaluation of pharmacological active compounds (Feng *et al*, 2009). As ERBB4 is a member of the ERBB family, we determined spironolactone's antagonistic effects on dimer formation of other ERBB receptors (Fig 3E). As indicated by $IC_{50}$ values obtained from dose–response assays (Figs 3F and EV4A–I), spironolactone preferentially inhibits formation of ERBB4 homodimers (at 1.75 μM). Spironolactone antagonizes also other ERBB combinations, albeit with less efficacy, and efficiently inhibits EGFR homodimer formation when stimulated by EGF (at 1.74 μM) (Figs 3F and EV4J). Collectively, these data suggest that spironolactone acts as a pan-ERBB family inhibitor, which preferentially inhibits dimer formation involving EGFR or ERBB4. In adult cortical tissues of mice, however, EGFR is not detectably expressed (Appendix Fig S2), suggesting that ERBB4 is the major target of the ERBB family for spironolactone in the brain.

## Spironolactone reverts ERBB4 hyperphosphorylation

Dose–response assays indicated that spironolactone reduces the phosphorylation-dependent recruitment of PIK3R1 by activated ERBB4 (Figs 2C and 3A), prompting us to examine the phosphorylation levels reverted by spironolactone biochemically. EGFld treatment of human T-47D cells that endogenously express *ERBB4* induced hyperphosphorylation at Tyr1056 and Tyr1284 of ERBB4. Addition of lapatinib completely reverted Tyr1056 and Tyr1284 phosphorylation, whereas treatment with spironolactone reduced phosphorylation to intermediate levels (Fig 4A and B) Likewise, spironolactone antagonized EGFld-mediated hyperphosphorylation of transfected human ERBB4 in PC12 cells that were used in the screen (Fig EV3D). To translate our findings into a potential therapeutic rationale for SZ, we utilized a transgenic mouse model, in which *Nrg1 type III* is overexpressed under the control of the neuronal Thy1.2 promoter (referred to as *Nrg1*-tg) and causes hyperphosphorylation of ERBB4 receptors in the prefrontal cortex (Velanac *et al*, 2012). We tested whether phospho-ERBB4 levels were also regulated by chronic spironolactone treatment in *Nrg1*-tg mice and injected the drug for 21 consecutive days before sacrificing the mice for biochemical analysis (Fig 4C). In lysates from mouse prefrontal cortex, phospho-Erbb4 levels were efficiently visualized using the p-ERBB4-Y1284 antibody (Fig 4D and E). Notably, addition of spironolactone resulted in a robust reduction of NRG1-induced ERBB4 hyperphosphorylation. Downstream signaling effects, as assessed by pERK and pAKT, were neither modulated by Nrg1 overexpression, nor by spironolactone treatment. Next, we tested whether LIM kinase 1 (LIMK1) displays regulated phosphorylation levels upon spironolactone treatment. LIMK1 is a non-receptor protein serine/threonine kinase implicated in cytoskeleton dynamics and the regulation of synaptic spine morphology and function (Meng *et al*, 2002, 2004) and has been associated with

NRG1 signaling (Yin *et al*, 2013). Notably, phospho-Limk1 levels were slightly upregulated in wt and markedly upregulated in spironolactone-treated *Nrg1*-tg mice, suggesting that LIMK1 activity integrates, at least in part, spironolactone's inhibitory effect (Fig 4D and E). Taken together, these results indicate that spironolactone serves as an inhibitor of NRG1-mediated ERBB4 signaling.

Since NRG1-ERBB4 signaling has been shown to modulate inhibitory neurotransmission (Yin *et al*, 2013; Agarwal *et al*, 2014; Mei & Nave, 2014), we tested the impact of spironolactone treatment on synaptic transmission in acute slices prepared from prefrontal cortex. When we measured spontaneous inhibitory postsynaptic currents (sIPSCs) at layer II/III pyramidal neurons, canrenone (10 μM) showed no significant effect on sIPSC frequencies and amplitudes (Fig 4F). In contrast, administration of spironolactone (10 μM) caused an increase of sIPSC frequency ($n = 12$; $P < 0.05$) and amplitude ($n = 12$, $P < 0.05$) shortly after bath application (Fig 4G). Next, evoked IPSCs were measured in pyramidal neurons in layer II/III of prelimbic cortex after stimulating in layer I. To distinguish between MR and ERBB4-mediated effects by spironolactone, these experiments were performed in the presence of canrenone (10 μM), which is a more potent MR antagonist relative to spironolactone. Under these conditions, spironolactone (5 μM) significantly increased the eIPSC amplitude (Fig 4H), a finding also increased by the pan-ERBB family inhibitor compound lapatinib as control (5 μM) (Fig 4I). Together, these findings are consistent with the hypothesis that spironolactone modulates GABAergic neurotransmission via ERBB4.

## Chronic spironolactone treatment ameliorates SZ-relevant behavioral endophenotypes in Nrg1-tg mice

*Nrg1*-tg mice exhibit SZ-relevant behavioral abnormalities, including deficits in prepulse inhibition (PPI) (Agarwal *et al*, 2014), an operational measure of sensorimotor gating. In a pilot experiment, we performed a two-arm study, in which *Nrg1*-tg and wt mice were tested for PPI before and after chronic spironolactone treatment as above (Fig EV5A). Before treatment, *Nrg1*-tg mice displayed PPI deficits (Fig EV5B), in line with our previous findings (Agarwal *et al*, 2014). Spironolactone treatment significantly improved PPI in *Nrg1*-tg mice (Fig EV5C), but had no effect in wt controls, suggesting that spironolactone modulates behavioral deficits of enhanced NRG1-ERBB4 signaling (Fig EV5D).

Based on these results, we performed a four-arm study, in which an independent cohort of *Nrg1*-tg and wt mice was tested following spironolactone or vehicle treatment (Fig 5A). In this study, we assessed a battery of behavioral domains with relevance for SZ, such as motor activity, curiosity, light–dark preference, working memory, motivation, PPI, fear memory, and pain sensitivity.

Vehicle-treated *Nrg1*-tg mice covered longer distances in the open-field arena than vehicle-treated wt controls (Fig 5B and C). This locomotor hyperactivity, however, was reverted by spironolactone (Fig 5B and D). Further, *Nrg1*-tg mice showed, increased anxiety, paralleled by higher frequency of defecation and urination during the open-field test (Fig EV5E–G), supporting previous observations (Agarwal *et al*, 2014). When testing for light–dark preference, spironolactone-treated *Nrg1*-tg spent more time in the light compartment. This suggests an anxiolytic rather than a sedative effect of spironolactone (Fig 5E), as *Nrg1*-tg animals and wt controls

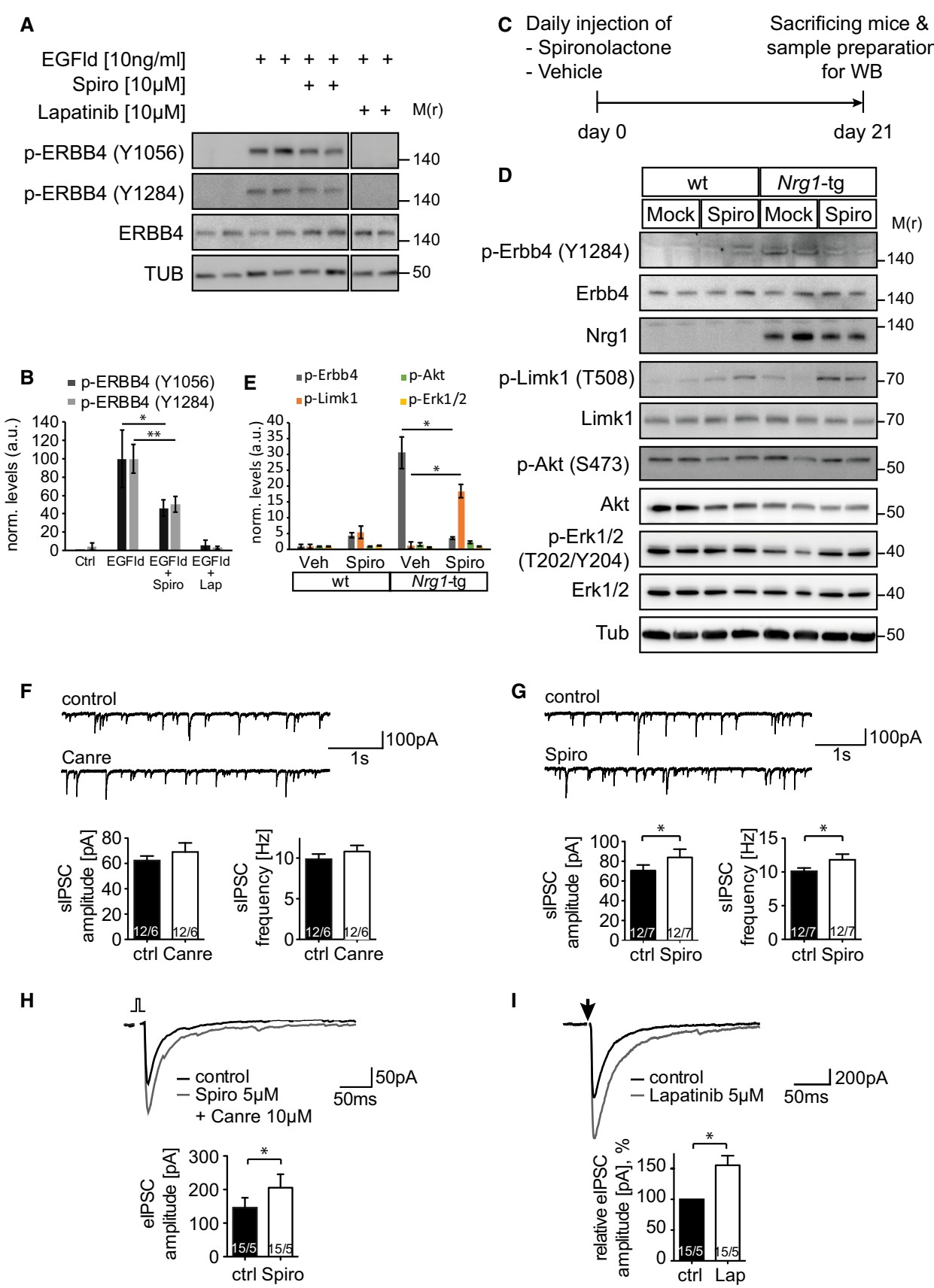

Figure 4.

◄

**Figure 4.  Spironolactone antagonizes ERBB4 phosphorylation both in *in vitro* and *in vivo*.**

A   Spironolactone reduces ERBB4 levels. T-47D cells were stimulated with 10 ng/ml EGFld, 10 µM lapatinib and 10 µM spironolactone for 5 min as indicated. Cell lysates were probed for ERBB4 phosphorylation levels at Tyr1056 and Tyr1284.

B   Quantification of band intensities for phospho-ERBB4 levels ($n$ = 4 per condition) shown in (A) using ImageJ. Phosphorylation levels are normalized to protein levels of ERBB4. Data are shown as mean, and error bars represent SD; *t*-test, with *$P$ = 0.0356 for p-ERBB4 (Y1056), and **$P$ = 0.0079 for p-ERBB4 (Y1284).

C   Experimental design for Western blot analysis shown in (D) and (E). *Nrg1*-tg and wt animals were treated daily with spironolactone (50 mg/kg, s.c.) or vehicle ($n$ = 2 per genotype and per treatment) for 21 days.

D   Spironolactone reduces phospho-Erbb4 levels in *Nrg1*-tg mice. Mice were treated with spironolactone for 21 days and sacrificed for Western blot analysis. Lysates were probed with indicated antibodies.

E   Quantification of band intensities for phospho-Erbb4 and phospho-Limk1 levels ($n$ = 2 per condition) shown in (D) using ImageJ. Phosphorylation levels are normalized to protein levels of Erbb4 and Limk1. Data are shown as mean, and error bars represent SD; *t*-test, with *$P$ = 0.0330 for p-Erbb4, and *$P$ = 0.0201 for p-Limk1.

F   Canrenone (applied as 10 µM) showed no effects on frequencies and amplitudes of sIPSCs in pyramidal neurons of the prefrontal cortex (PFC). Representative traces of sIPSCs (upper) and a histogram of mean sIPSC (lower) are shown for both before and after addition of canrenone.

G   Spironolactone (applied as 10 µM) significantly increases frequencies ($n$ = 12; *$P$ = 0.0454) and amplitudes ($n$ = 12, *$P$ = 0.0478) of sIPSCs in pyramidal neurons of PFC. Representative traces of sIPSCs (upper) and a histogram of mean sIPSC (lower) are shown for both before and after addition of spironolactone.

H   Spironolactone increases amplitudes of evoked IPSCs in pyramidal neurons of PFC ($n$ = 15, *$P$ = 0.0286). Spironolactone (5 µM) was applied in the presence of canrenone (10 µM).

I   Lapatinib increases amplitudes of evoked IPSCs in pyramidal neurons of PFC ($n$ = 15, *$P$ = 0.0124). Lapatinib was applied at 5 µM. Arrow indicates an evoked IPSC stimulus.

Data information: For (H, I), sample recording (upper) and a histogram of averaged eIPSC (lower) is shown for both before and after drug application. For (F–I), the numbers displayed inside the histogram bars indicate the number of recorded slices/number of animals. Data are shown as mean, and error bars represent SD (for B, E) and SEM (for F–I); paired *t*-test, with *$P \leq 0.05$; Spiro, spironolactone; Canre, canrenone; Lap, lapatinib.

Source data are available online for this figure.

displayed a similar transition activity in the light–dark test (Fig EV5H). In contrast, *Nrg1*-tg mice showed an increased activity in the tail suspension test (Fig EV5I). Working memory was assessed in the Y-maze test. *Nrg1*-tg mice performed significantly less alterations than wt controls, suggesting an impaired working memory performance in transgenics (Fig 5F). Notably, spironolactone treatment rescued these deficits (Fig 5F), without influencing the activity in the Y-maze (Fig EV5J). However, the number of choices was higher in transgenics, supporting their hyperactivity phenotype observed in the open-field test (Fig 5B and C).

In the contextual fear memory test, spironolactone-treated *Nrg1*-tg mice displayed a non-significant reduction in freezing (Fig EV5K), which was paralleled by significantly decreased levels of pain sensitivity as assessed in the hot plate test (Fig EV5L). Cue memory testing revealed neither genotype nor treatment-dependent alterations (Fig EV5K). Moreover, spironolactone treatment significantly enhanced PPI in *Nrg1*-tg (Fig 5G), but not in wt mice (Fig EV5M), replicating the data obtained from the pilot experiment (Fig EV5D).

As Nrg1 is a critical regulator for brain development, we aimed to exclude age-related consequences on the behavioral effects observed using a covariate analysis, which links age with test performance. To do this, we grouped mice into juvenile (8–11 weeks) and adult (12–16 weeks) categories, and ANCOVA with the covariate age did not reveal an effect on the genotype-dependent treatment response (Appendix Fig S3).

Taken together, chronic spironolactone treatment alleviates hyperactivity, PPI and working memory deficits in *Nrg1*-tg mice, findings that are paralleled by a reduction in Erbb4 hyperphosphorylation levels in these mice.

## Discussion

To obtain repurposed drugs for schizophrenia, we have developed a co-culture assay system mimicking several proximal aspects of

NRG1-ERBB4 signaling (Citri & Yarden, 2006; Mei & Xiong, 2008). Its successful application in a repurposing screen with clinical substances resulted in the identification of the MR antagonist spironolactone as a potent ERBB4 inhibitor. The efficacy of spironolactone as novel inhibitor of NRG1-ERBB4 signaling was validated in heterologous cells with an endogenous expression of human ERBB4 and *in vivo* using transgenic mice, which model NRG1 overexpression and ERBB4 hyperphosphorylation linked to several endophenotypes with relevance for SZ (Agarwal *et al*, 2014). We thus provide a pharmacological proof-of-principle for targeting NRG1-ERBB4 signaling in the context of SZ and exploited the opportunity to repurpose a clinical compound, a strategy that was strongly demanded in the last years for mental diseases (Insel, 2012). In addition, a recent study suggested to fast track all SZ susceptibility genes, which encode potential targets for approved drugs, for repurposing (Lencz & Malhotra, 2015). Therefore, our multilevel approach targeting NRG1-ERBB4 signaling that identified a hidden mode-of-action of spironolactone antagonizing ERBB4 activity strongly supports this approach. As spironolactone is a clinically safe and available substance, it immediately qualifies for therapeutic intervention trials. Finally, the co-culture assay is qualified for high-throughput conditions under industry quality standards and will allow the exploratory screen of large exploratory compound libraries.

Spironolactone inhibited the association between ERBB4 and PIK3R1 in the split TEV-based co-culture assay, with an $IC_{50}$ value at approximately 1 µM. Our data suggest that spironolactone also targets other ERBB family members, however, with a preference for ERBB4. Our biochemical analysis suggests that spironolactone shows an intermediate efficacy of ERBB4 inhibition. Notably, fine tuning of excitation and inhibition between excitatory projection neurons expressing NRG1 and inhibitory parvalbumin-positive interneurons expressing ERBB4 is thought to be a critical determinant of endophenotypes observed in gain- and loss-of-function mouse models (Chen *et al*, 2010; Yin *et al*, 2013; Agarwal *et al*, 2014). Therefore, moderate changes in NRG1/ERBB4 activity may be desired to achieve rebalanced signaling levels under pathological

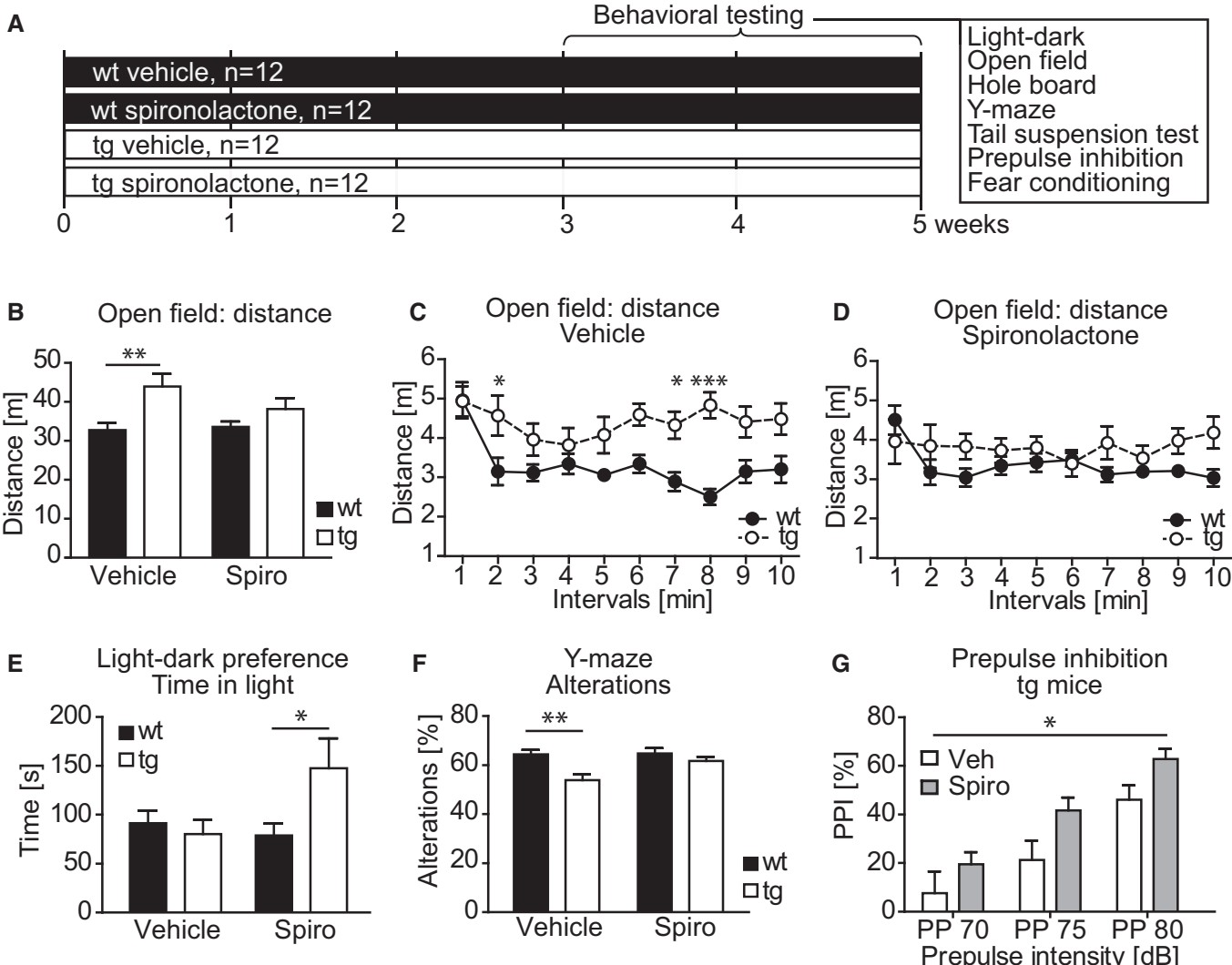

**Figure 5. Chronic spironolactone treatment ameliorates deficits of behavioral endophenotypes in *Nrg1*-tg mice.**

A   Experimental design. *Nrg1*-tg and wt animals were treated daily with spironolactone (50 mg/kg, s.c.) or vehicle (n = 12 per genotype and per treatment) for 3 weeks, followed by behavioral phenotyping using the tests as indicated. Spironolactone or vehicle treatment was continued throughout the phenotyping phase.

B   *Nrg1*-tg mice travelled longer distances in the open-field arena (effect of genotype $F_{1,44}$ = 10.53; P = 0.0022; two-way ANOVA). Bonferroni *post hoc* analysis revealed a significant genotype-dependent difference between vehicle-treated groups (**P = 0.0044) but not in spironolactone-treated groups (P = 0.3783). Genotype differences were abolished upon spironolactone treatment.

C   When vehicle-treated animals were analyzed in 1-min intervals, transgenic mice showed an increased activity throughout the entire test (effect of genotype $F_{1,22}$ = 9.27; P = 0.0060; two-way ANOVA), most prominent in intervals 2, 7, and 8 (*P = 0.0427, *P = 0.0385, and ***P = 0.000036, respectively, Bonferroni *post hoc* test).

D   There was no significant difference between the genotypes when treated with spironolactone (effect of genotype $F_{1,22}$ = 2.19; P = 0.1535; two-way ANOVA). However, the interaction of genotype and treatment was significant ($F_{9,198}$ = 1.94; P = 0.0481; two-way ANOVA).

E   *Nrg1*-tg mice treated with spironolactone spent more time in the light compartment during the light–dark test (interaction gene × treatment $F_{1,41}$ = 4.90; P = 0.0324; two-way ANOVA, and *P = 0.0219, Bonferroni *post hoc* test).

F   In the Y-maze test, transgenic mice performed less alterations (effect of genotype $F_{1,44}$ = 11.50; P = 0.0015; two-way ANOVA). The Bonferroni test confirmed this phenotype in vehicle-treated groups (**P = 0.0011), but not in spironolactone-treated animals (P = 0.5950). Spironolactone treatment had a significant effect on the number of alterations ($F_{1,44}$ = 4.12; P = 0.0484; two-way ANOVA).

G   Spironolactone treatment significantly enhanced PPI in *Nrg1*-tg mice (effect of treatment $F_{1,21}$ = 5.07; *P = 0.0325; two-way repeated-measures ANOVA).

Data information: Data are shown as mean, and error bars represent SEM. Spiro, spironolactone; Veh, vehicle. n = 12 per genotype and treatment with an exception of (E) (*Nrg1*-tg vehicle, n = 11; *Nrg1*-tg Spiro, n = 10; wt vehicle, n = 12; wt Spiro, n = 12) and (G) (*Nrg1*-tg vehicle, n = 11; *Nrg1*-tg Spiro, n = 12).

conditions. We found that the chronic administration of spironolactone to *Nrg1*-tg mice reverted Erbb4 hyperphosphorylation and largely rescued hyperactivity (considered as a surrogate marker for positive symptoms in SZ), PPI deficits and working memory impairments. Importantly, behavior in wild-type mice remained largely unaffected by chronic spironolactone treatment. Spironolactone, but

not canrenone, enhanced inhibitory neurotransmission when applied acutely to cortical slices of wild-type mice, suggesting an ERBB4-mediated mechanism. Likewise, the ERBB4 kinase inhibitor lapatinib caused similarly increased IPSCs within the same experimental model. Increased amplitudes of mIPSCs have also been observed in conditional Nrg1 loss-of-function mutants, most likely as a consequence of reduced activity of ERBB4 in inhibitory neurons (Agarwal *et al*, 2014). In conditional ERBB4 mutants, however, mIPSC frequencies were reduced in the hippocampus (Fazzari *et al*, 2010). A recent study reports that NRG2, a close relative of NRG1, is expressed in inhibitory interneurons and activates ERBB4 cell-autonomously, causing a downregulation of NMDA receptor activity in these cells (Vullhorst *et al*, 2015). In such a scenario, inhibition of ERBB4 activity may indeed increase IPSCs, a hypothesis fitting to our observations, for both spironolactone and lapatinib control treatments. Overall, these findings indicate that altered NRG1/ERBB4 signaling modulates inhibitory signaling, although different adaptations may prevail in different brain regions and genetic models as well as pharmacological treatments.

Our biochemical analysis implicates LIMK1 signaling, but not ERK1/2 nor AKT1 as potential downstream effectors of spironolactone treatment in *Nrg1*-tg mice. As a non-receptor protein serine/threonine kinase, LIMK regulates synaptic spine morphology and function by modulating cytoskeleton dynamics (Meng *et al*, 2002, 2004; Bennett, 2011). Further, LIMK1 has been linked to NRG1 signaling and SZ-relevant endophenotypes in a *Nrg1*-tg mouse model (Yin *et al*, 2013). We show that phospho-LIMK1 levels were upregulated in *Nrg1*-tg mice treated with spironolactone suggesting that LIMK1 activity may possibly integrate spironolactone's inhibitory effect by promoting spine enlargement, and thus synapse formation, through controlling actin cytoskeleton dynamics. *Nrg1*-tg animals display subtle structural changes related to spine morphology, that is, the number of bifurcated spines is increased (Agarwal *et al*, 2014). Therefore, it might be possible that spironolactone treatment reverts this structural endophenotype. Nonetheless, the increased levels of p-LIMK1 rather favor a mechanism compensating for the structural changes in *Nrg1*-tg mice, which may underlie network disturbances in these animals, by stimulating structural plasticity via increased LIMK1 activity. To further explore the mode-of-action of spironolactone in the future, its impact on structural plasticity should be addressed in additional studies.

Spironolactone has been developed as MR antagonist and was clinically applied for decades as a potent and safe diuretic (Ogden *et al*, 1961). As brain-expressed corticoid receptors are implicated in modulating the stress response, spironolactone treatment has been tested in the context of depression and was shown to increase motivation and curiosity in mice (Wu *et al*, 2012). Moreover, anxiety was partially improved in a small group of patients suffering from bipolar disorder (Juruena *et al*, 2009) in good agreement with our finding that spironolactone affects anxiety-related behavior in *Nrg1*-tg mice. Acute spironolactone administration to healthy human volunteers, however, reduced memory retrieval (Zhou *et al*, 2011; Rimmele *et al*, 2013) and reportedly impaired recent fear memory formation in mice (Zhou *et al*, 2011). Upon chronic administration of spironolactone, however, we could not observe any detrimental effects on cognitive performance in fear and working memory tests in wild-type mice. However, spironolactone-treated *Nrg1*-tg mice displayed a slightly reduced level of fear memory, which may be partially dependent on anxiolytic actions of spironolactone or decreased pain sensitivity in transgenic mice. Nonetheless, working memory deficits of *Nrg1*-tg mice were rescued upon spironolactone treatment.

Structure-function analysis using spironolactone metabolites and second-generation analogues revealed that the intact structure of spironolactone is paramount for inhibiting ERBB4 signaling activity. We speculate that other structural modifications of spironolactone may improve its selectivity for ERBB4 binding and concomitant inhibition. Spironolactone may therefore also serve as a template for a lead optimization process, which could produce a new molecular entity with improved characteristics to inhibit ERBB4 signaling and avoiding potentially adverse effects on the MR. Nonetheless, given our observations and the safety profile of spironolactone, a clinical study might be warranted to assess the chronic effects of spironolactone treatment in SZ patients.

# Materials and Methods

### Plasmids

Gateway recombination cloning (Life Technologies) was applied for generating plasmids. Each ORF cloned was PCR-amplified using the Pwo proofreading DNA polymerase (Roche) and BP-recombined into the pDONR/Zeo plasmid (Life Technologies) to yield an entry vector, which was control-digested using BsrGI to release the insert, and sequence-verified. The following human ORFs were cloned using cDNA image clones ordered from Source BioScience: ERBB4 transcript variant JM-a/CYT-1 (Accession: BC112199), PIK3R1 (BC094795), GRB2 transcript variant 1 (BC000631), SHC1 transcript variant 2 (BC014158), STAT5A (BC027036), and SRC transcription variant 1 (BC051270). For EGFR, ERBB2, and ERBB3, entry vectors were obtained from the human CCSB kinase collection available from Addgene (Johannessen *et al*, 2010). Entry vectors were LR-recombined into split TEV destination vectors [either pcDNA_attR1-ORF-attR2-NTEV-tevS-GV-2xHA_DEST or pTag4C_attR1-ORF-attR2-CTEV-2xHA_DEST, plasmids were described in detail before (Wehr *et al*, 2006)] to yield expression vectors, which were control-digested using BsrGI. The plasmids for Nrg1 type I β1a and rat Nrg1 type III β1a have been described before (Wehr *et al*, 2006).

The oligonucleotides used for cloning are shown in the Appendix Table S2.

### Cell culture

PC12 Tet-Off cells (Clontech, 631134, termed PC12 cells for simplicity) and their derivatives stably expressing Nrg1 were cultured in low glucose DMEM medium (1 g/l, Lonza) supplemented with 10% FCS, 5% HS, 50 μg penicillin, 50 μg streptomycin and GlutaMAX (Life Technologies) at 37°C, and 5% $CO_2$. T-47D cells (ATCC:HTB-133) were grown RPMI-1640 (Life Technologies) supplemented with 0.2 units/ml bovine insulin (Sigma), 10% FCS, 50 μg penicillin, 50 μg streptomycin and GlutaMAX (all Life Technologies) at 37°C, and 5% $CO_2$. PC12 cells were grown on poly-L-lysine-coated surfaces for both maintenance and experiments; T-47D cells were only grown on poly-L-lysine-coated surfaces for experiments. Cell lines were negative for mycoplasma contamination.

**Generation of PC12 cells stably expressing Nrg1**

1 Mio PC12 cells were either transfected with 10 µg of a Nrg1 type I β1a plasmid or Nrg1 type III β1a plasmid using Lipofectamine 2000. Following an initial expression of 24 h, 400 µg/ml G418 was applied to select stable clones as each Nrg1 plasmid harbors a neomycin resistance gene for selection in mammalian cells. After 2 weeks of culturing, visible PC12 cell clones were transferred into a single well of a 24-well plate. Following to a recovery and expansion phase, stable Nrg1 expression was validated in a split TEV-based ERBB4-PIK3R1 co-culture assay. Positive clones were also verified by Western blot analysis.

**Protein lysates and Western blotting**

PC12 cells were transfected with indicated plasmids using Lipofectamine 2000 (Life Technologies). After 24 h of expression, cells were treated as indicated and lysed in a 1% Triton-X lysis buffer (50 mM Tris [pH 7.5], 150 mM NaCl, 1% Triton X-100, 1 mM EGTA) supplemented with 10 mM NaF, 1 mM $Na_2VO_4$, 1 mM $ZnCl_2$ 4.5 mM $Na_4P_2O_7$, as phosphatase inhibitors, and the complete protease inhibitor cocktail (Roche). Lysates from T-47D cells were analyzed for endogenous proteins only. For the analysis of cytosolic proteins, cell extracts were spun for 10 min at 4°C at 17,000 $g$.

For the biochemical analysis of spironolactone-treated mice (for a precise description of the injection paradigm, see subheading "Mouse behavior analysis", "Spironolactone treatment"), vehicle control or spironolactone was subcutaneously injected daily for 21 days into age-matched (11–13 weeks) male mice prior to preparation of the mouse prefrontal cortex ($n = 2$ per genotype and treatment). For the generation of lysates, the isolated tissue was immediately placed into cooled (4°C) sucrose buffer (320 mM sucrose, 10 mM Tris–HCl, 1 mM $NaHCO_3$, 1 mM $MgCl_2$, supplemented with 10 mM NaF, 1 mM $Na_2VO_4$, 1 mM $ZnCl_2$, 4.5 mM $Na_4P_2O_7$ as phosphatase inhibitors, and the complete protease inhibitor cocktail (Roche)), homogenized using an ultra-turrax (IKA GmbH, Staufen, Germany), sonicated (3 pulses for 10 s), and denatured for 10 min at 70°C in LDS sample buffer.

Protein gels were run using the Mini-PROTEAN Tetra Electrophoresis System (Bio-Rad), and gels were blotted using the Mini-PROTEAN Tetra Electrophoresis System (Bio-Rad). Detection of proteins was performed by Western blot analysis using chemiluminescence (Western Lightning® Plus-ECL, PerkinElmer). Western blots were probed with antibodies at dilutions as shown in the Appendix Table S3. Each blot was replicated two times. Western blots were densitometrically quantified using ImageJ following the protocol openly accessible at lukemiller.org (http://lukemiller.org/index.php/2010/11/analyzing-gels-and-western-blots-with-image-j/).

**Immunofluorescence staining of PC12 cells**

1 Mio PC12 cells were plated per well onto poly-L-lysine (PLL)-coated coverslips in a 6-well plate at day 1. At day 2, cells were transfected with ERBB4-NTEV-tevs-GV or ERBB4_1-685-NTEV-tevS-GV. At day 3, cells were gently washed twice by adding and removing 1× TBS (50 µl per coverslip), fixed in cold 4% PFA for 10 min,

washed twice with 1× TBS, and permeabilized in TBS/0.1% Triton X-100 for 5 min. Then, cells were washed again three times in 1× TBS and blocked in blocking buffer (3% BSA, 0.1% Triton X-100 in 1× TBS) for 1 h at room temperature. Primary antibodies diluted in blocking buffer and cells were incubated for 1 h at room temperature. Following three washes in TBS, cells were incubated with a secondary antibody (Alexa 594 anti-rat, 1:500, Abcam, ab150160) diluted in blocking buffer for 1 h at room temperature. Coverslips were washed three times in 1× PBS, once quickly dipped into ddH$_2$O to remove traces of salt, mounted on microscope slides, and sealed with ProLong Gold Antifade Mountant with Dapi (ThermoFisher Scientific, P36935). Slides were stored at 4°C before imaged on a Zeiss Observer Z.1 microscope.

**Compound screening and validation**

*Cell-based split TEV assay to monitor ERBB4 activity*
The split TEV method is based on the functional complementation of two previously inactive TEV protease fragments denoted NTEV and CTEV fused to interacting proteins. It has been shown to robustly and sensitively quantify protein–protein interactions and receptor activities, as proven before for the ERBB4 receptor and the regulatory adapter subunits of the PI3K, PIK3R1, and PIK3R2 (Wehr *et al*, 2006, 2008). Recently, the split TEV method was also successfully applied to genomewide RNAi screening in *Drosophila* cell culture, supporting its applicability to high-throughput applications (Wehr *et al*, 2013).

For our HTS-compatible split TEV assay approach, human full-length ERBB4-Cyt1 was fused to the NTEV fragment, a TEV protease cleavage site (tevS) and the artificial co-transcriptional activator Gal4-VP16 (ERBB4-NTEV-tevS-GV); human PIK3R1 was fused to the CTEV fragment (PIK3R1-CTEV) (Fig 1A). Upon ERBB4 activation, PIK3R1 is recruited to the receptor resulting in a reconstituted protease activity that cleaves off GV. In turn, released GV translocates to the nucleus and binds to upstream activating sequences (UAS) to activate the transcription of a firefly luciferase reporter gene (Fig 1A). A constitutively expressed *Renilla* luciferase driven under the control of the human thymidine kinase (TK) promoter was used as control to address off-target effects related to toxicity.

*Compound library*
For small molecule screening, the NIH-NCC Clinical Collection library (sets NCC-003 and NCC-201) was used containing 727 small molecules that are FDA-approved and have a history in clinical applications (www.nihclinicalcollection.com). A Hamilton Labstar robot connected to 37 and 4°C incubators for cell incubation and compound storage and application (Cytomat automated incubator, ThermoScientific) and to a luciferase reader (Berthold Technologies) was used to automatically perform the screening. Batch 1 (compounds 1–320) was run in quadruplets, batch 2 (compounds 321–727) in triplicates. Each batch was screened three times.

*Transfection of cells*
To equally transfect large amounts of cells, PC12 cells were transfected with the split TEV assay components in solution. For one 96-well plate, $4 \times 10^6$ PC12 cells were harvested and diluted in 5 ml assay medium (phenol red-free DMEM (low glucose, Life

Technologies), 10% FCS, 5% HS, no antibiotics). The split TEV assay plasmids (2 μg pcDNA3_ERBB4-NTEV-tevS-GV-2xHA, 2 μg pTag4C_PIK3R1-CTEV-2xHA, 2 μg p5xUAS_firefly luciferase, 2 μg pTK_*Renilla* luciferase, and 0.5 μg pECFP-C1 for examining transfection efficiency) were diluted in 2.5 ml Opti-MEM (Life Technologies) and vortexed. In parallel, 20 μl of the transfection reagent Lipofectamine 2000 was diluted in 2.5 ml Opti-MEM and vortexed. Both Opti-MEM aliquots were mixed, vortexed, and incubated for 20 min at room temperature, followed by carefully mixing the DNA/Lipofectamine/Opti-MEM solution with the PC12 cells and incubating the cell suspension at 37°C and 5% $CO_2$ for 2 h without shaking.

### Plating the cells

For plating of one 96-well plate, 10 ml suspension, containing the $4 \times 10^6$ in solution-transfected cells, was placed in the bubble paddle reservoir of the Hamilton Cellstar robot. 100 μl was seeded per 96-well using the 96-tip pipetting head. The homogeneity of the cell suspension was guaranteed over time by mild stirring using the paddling device inside the reservoir. For five plates each, 50 ml of additional cell suspension was used to allow for losses of inaccessible volume. After seeding, plates were transferred and stored in the Cytomat device at 37°C and 5% $CO_2$.

### Addition of compounds

The cells were allowed to express the plasmids for 24 h before compounds were added. Proper expression and transfection efficiency were verified by ECFP expression on a clear control plate. The compounds were applied in a final concentration of 10 μM using DMSO as diluent. Sixteen positions per 96-well plate (i.e., columns A and H) were reserved for controls; in detail, four wells each were taken for positive controls (stimulated with 10 ng/ml EGFld (Reprokine, RKQ02297) in DMSO, 96-well positions A1 to D1), baseline controls (DMSO only, 96-well positions E1 to H1), negative controls I (100 nM CI-1033 (Canertinib dihydrochloride, Axon, 1433) in DMSO, 96-well positions A12-D12), and negative controls II (10 μM lapatinib (Lapatinib ditosylate, Axon, 1395) in DMSO, 96-well positions E12 to H12). Thirty minutes later, 10,000 Nrg1-expressing PC12 cells in 100 μl assay medium were seeded on top. 24 h after addition of the compounds, the cells were lysed using 40 μl Passive Lysis buffer (Promega) and subjected to a Dual Luciferase Assay (Promega) according to the manufacturer's instructions. The data were analyzed in R Bioconductor using the package cellHTS2 (http://www.bioconductor.org/packages/devel/bioc/html/cellHTS2.html), assessed using the z-score, and visualized using the program Mondrian (http://stats.math.uni-augsburg.de/mondrian/).

### Dose–response luciferase assays for validation

Individually re-screened candidates were validated using a dose–response assay. PC12 cells were batch-transfected as described in the section "Transfection of cells", manually plated, and incubated for 24 h at 37°C and 5% $CO_2$. Candidate small molecules were prepared in a series of dilutions using DMSO as diluent and ranging from 0.0001 to 100 μM at final concentrations, thus covering at least five orders of magnitude. Candidate dilutions were added, followed by the addition of 10,000 Nrg1-expressing cells in 100 μl volume 30 min later. Cells were lysed in 40 μl Passive Lysis Buffer and

analyzed in a Dual Luciferase Assay. Data were analyzed in Excel and GraphPad Prism. For single-culture assays that used EGFld as stimulus, 100 μl assay medium containing EGFld (f.c. 10 ng/ml) was administered. The following candidates were analyzed in dose–response assays: spironolactone (Sigma-Aldrich, S3378), eplerenone (Sigma-Aldrich, E6657), canrenone (Santa Cruz Biotechnology, sc-205616), 7α-thiomethyl-spironolactone (Santa Cruz Biotechnology, sc-207187). Dose–response assays were run in six replicates per concentration and repeated at least two times. Data are shown as mean, and error bars represent SEM.

### Electrophysiology

300-μm-thick transverse slices comprising the medial prefrontal cortex (mPFC) area were prepared from 7- to 8-week-old C57Bl/6 mice as described previously (Teng *et al*, 2013; Agarwal *et al*, 2014). All recordings were performed in cortical layer V projection neurons. The bath solution consisted of oxygenated artificial cerebrospinal fluid (ACSF) containing NaCl 126 mM, KCl 3 mM, $NaH_2PO_4$ 1.25 mM, $MgSO_4$ 1 mM, $NaHCO_3$ 26 mM, $CaCl_2$ 2 mM, glucose 10 mM, and the pH 7.2 adjusted with NaOH. Spontaneous GABAergic inhibitory postsynaptic currents (sIPSC) and evoked GABAergic inhibitory postsynaptic currents (eIPSC) were recorded at a holding potential of −70 mV in the presence of 6-cyano-7-nitroquinoxaline-2,3-dione (CNQX, 10 μM) and DL-2-amino-5-phosphonovaleric acid (D-APV, 50 μM). Glass pipettes were filled with the solution containing: KCl 140 mM, $MgCl_2$ 2 mM, $CaCl_2$ 1 mM, $Na_2GTP$ 0.5 mM, $Na_2ATP$ 4 mM, EGTA 10 mM, HEPES 10 mM. Patches with a series resistance of > 25 MΩ, a membrane resistance of < 0.8 GΩ, or leak currents of > 150 pA were excluded. Biphasic rectangular electric pulses (10 ms, 100–300 μA) were applied to layer I of the prelimbic area of mPFC through another glass pipette filled with ACSF. Synaptic currents were acquired at 20 kHz and filtered at 6 kHz, with a Digidata 1440 ADC-converter coupled to a Multiclamp 700B amplifier (Molecular Devices, USA). Data acquisition was performed using pClamp10 software (Molecular Devices). MiniAnalysis 6.0.9 (Synaptosoft Inc., Decatur, USA) and pClamp10.1 were used for amplitude and frequency analysis of sIPSCs and eIPSCs, respectively. Paired Student's *t*-test was used for statistical analysis.

### Mouse behavior analysis

For behavioral testing, age-matched male mice (8–16 weeks) on C57Bl/6 background that constitutively overexpress the 2xHA-tagged Nrg1 type III β1a isoform (*Nrg1*-tg) under the control of the mouse Thy1.2 promoter (Velanac *et al*, 2012) and their wild-type (wt) littermates as controls were used. Animals were group-housed in the same ventilated sound-attenuated rooms under a 12-h light/12-h dark schedule (lights on at 8:00 am) at an ambient temperature of 21°C with food and water available *ad libitum*. One week prior to experiments, mice were separated into single cages and habituated to the experimental rooms. To minimize the influence of the circadian rhythm on drug actions, the treatment groups were analyzed at balanced time points during the light phase. The investigators for behavioral tests were blind to genotypes and/or spironolactone administration. All animal experiments were conducted in accordance with NIH principles of laboratory animal care and were

approved by the Government of Lower Saxony, Germany, in accordance with the German Animal Protection Law.

### Spironolactone treatment

50 mg of Spironolactone (Sigma-Aldrich) was initially dissolved in DMSO and suspended in 10 ml of 0.9% NaCl, 1% DMSO, and 0.002% Tween®20. Spironolactone (5 mg/ml) and a vehicle control (0.9% NaCl, 1% DMSO, and 0.002% Tween®20) were subcutaneously injected with a 50 mg/kg dose daily (e.g., corresponding to 0.3 ml injection volume of a mouse with 30 g weight) for 3 weeks prior to behavioral testing ($n = 12$ per genotype and per treatment). Treatment was continued throughout the behavioral analysis period. To avoid injection-induced stress prior to behavioral testing, mice were injected in the afternoon, after the entire cohort has completed the behavioral paradigm.

### Calculation of spironolactone dosage

The calculated daily dosage of 50 mg/kg/day spironolactone for mice is based on the following assumptions. Patients are routinely treated with 400 mg/day spironolactone (Aldactone 100, Riemser Pharma; spironolactone 100, Ratiopharm). Dosages of 50 to 100 mg/day were administered to patients in long-term treatments (Juruena *et al*, 2009). The dosage of 400 mg/80 kg patient body weight per day is equal to 5 mg/kg/day. Human doses are converted to mouse doses using the body surface area normalization method, which integrates various aspects of biological parameters including basal metabolism, blood volume, caloric expenditure, and oxygen utilization (Reagan-Shaw *et al*, 2008). For the calculation of the mouse dose (mg/kg), the human dose (mg/kg) is multiplied by the human $K_m$/mouse $K_m$, where the human $K_m = 37$ and the mouse $K_m = 3$. Therefore, mice should be treated with a 12-fold higher dose. The chosen dosage of 50 mg/kg/day is slightly below the calculated maximum dose of 61.7 mg/kg/day [(400 mg/80 kg)* (37/3)/day]. The $lC_{50}$ of spironolactone is > 1,000 mg/kg/day. Spironolactone is FDA-approved, used in patients for decades, and shows no major side effect in treated mice.

### Behavioral tests applied for mouse behavior analysis

**Open field and hole board** Spontaneous locomotor activity was verified in the open-field test using a Plexiglas box (45 × 45 × 55 cm). The same test arena was modified with a floor insert containing 16 symmetrically allocated holes for the hole board test. During a 10-min testing session, mouse behavior was monitored by infrared sensors and recorded by the ActiMot software (TSE, Bad Homburg, Germany). Levels of urination (scored in events) and defecation (scored as feci in events) were determined manually during the open-field test.

**Light–dark preference** The light–dark preference test was conducted in a plastic chamber divided into two compartments of same size, with one having black and the second one having transparent Plexiglas walls. A door-like opening in the center of the separating wall allowed transitions between both compartments. For testing, each mouse was placed into the light compartment facing away from the door and left undisturbed. The latency to enter the dark compartment, the time spent in the dark compartment, and the number of crossings between the compartments were monitored

for 5 min using the AnyMaze software. Mice that did not enter the dark compartment within 10 min were excluded from the experiment. After each session, the chambers were cleaned with 70% ethanol.

**Y-maze** The assessment of working memory was performed using an in-house made Y-shaped runway. Animals were placed individually into the Y-maze facing the wall and allowed to explore the maze for 10 min. The experiment was video recorded. The number of arm choices (as a measure of activity) and the percent of alterations (choices of a "novel" arm, i.e., when animals chose a different arm as before is regarded as a measure of working memory) were scored and analyzed. To avoid any olfactory cues, the apparatus was cleaned with 70% ethanol between animals.

**Tail suspension test** Mice were manually suspended upside down for 6 min by attaching them to a fixed rod using an adhesive tape positioned at the tip of the tail. The escape motivation of a mouse was measured as the time spent active, video recorded, and scored offline.

**Prepulse inhibition (PPI)** The startle response was measured using a two test cabinet (SR-LAB, San Diego Instruments) using a protocol as described in Brzózka *et al* (2010).

**Fear conditioning** Fear memory assessment that is measured by freezing behavior was performed using the Ugo Basile Fear Conditioning System (Varese, Italy). For conditioning, mice were placed into the animal box (furnished with a stainless shock grid floor and striped black-white walls) and positioned into an isolation cubicle equipped with a lamp, a loudspeaker and infrared camera. For conditioning, striped black-white walls were inserted into the animal box. The conditioning and fear memory assessment were performed as described in Brzózka *et al* (2010).

**Hot plate** Pain sensitivity was measured in the hot plate test. Animals were placed onto a metal plate preheated to 52°C. The latency to the first reaction (hind paw licking or jumping) was scored manually. Immediately after the first response, mice were placed onto another metal plate (not heated) to allow cooling their paws.

### Statistical analysis

Statistical significance was determined using Microsoft Excel, IBM SPSS Statistics v22 and GraphPad Prism 5.0 software. Data are presented as means ± SD or SEM as indicated ($n \geq 3$, for luciferase assays $n = 6$). For behavioral experiments, Student's *t*-tests were used for comparing two data samples. If the experimental setup required a paired data analysis, paired Student's *t*-test or paired Wilcoxon signed ranks test was used for comparing two normally or not-normally distributed data samples, respectively. Two-way ANOVA with Bonferroni *post hoc* test was used for the analysis of three or more samples. Two-way ANCOVA was used for the age-corrected analyses of open-field, Y-maze and light–dark preference tests. Repeated-measures ANOVA was used to analyze the effects of treatment in the PPI analyses. The robustness of cell-based assays was assessed using the Z' factor. Data from screening were analyzed

**The paper explained**

**Problem**

NRG1-ERBB4 signaling is a schizophrenia risk pathway in humans and altered signaling activity causes schizophrenia-relevant endophenotypes in transgenic mouse models. To date, no treatment options are available targeting this pathway in schizophrenic patients.

**Results**

Here, we have developed a NRG1-ERBB4 pathway-selective screening assay based on the split TEV technology to monitor activities of FDA-approved drugs for repurposing. The anti-mineralocorticoid spironolactone was identified as top candidate from the screen to antagonize ERBB4 receptor activity. Spironolactone's effect was biochemically validated both *in vitro* and *in vivo*, and it was found to improve schizophrenia-relevant behavioral deficits in a *Nrg1* transgenic mouse model.

**Impact**

We provide preclinical evidence for an approved drug that may immediately qualify for a clinical study in schizophrenic patients.

using the cellHTS2 package available for R and evaluated using the z-score.

**Expanded View** for this article is available online.

## Acknowledgements

We thank Elena Ciirdaeva, Nadia Gabellini, Monika Rübekeil, and Barbara Meisel for excellent technical support. M.C.W. was supported by the Deutsche Forschungsgemeinschaft (WE 5683/1-1). M.J.R. was supported by the Deutsche Forschungsgemeinschaft (Klinische Forschergruppe (KFO) 241: RO 4076/1-1 and PsyCourse: RO 4076/5-1). W.Z. was supported by the IZKF of the University of Münster Medical School (Zha3-005-14) and by the Deutsche Forschungsgemeinschaft (SFB TRR58 to W.Z.).

## Author contributions

Designed, performed, and analyzed the compound screen: MCW, WH, SPW, MJR; performed luciferase and biochemical validation experiments: MCW, WH, JPW, AH; performed immunocytochemistry: MCW, MCS-B; performed electrophysiological experiments: MK, MZ; designed behavioral analysis: MMB; performed and analyzed behavioral experiments: MMB, WH, TU, SP, MCW, MJR; supervised students and provided essential reagents: K-AN, PF, WZ, MHS; conceived the study and wrote the manuscript: MCW, MJR.

## Conflict of interest

Systasy Bioscience GmbH holds the patent for the split TEV technique (termed splitSENSOR technology at Systasy Bioscience GmbH). M.C. Wehr, S.P. Wichert, and M.J. Rossner are co-founders and shareholders of Systasy Bioscience GmbH, Munich, Germany.

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
