## [Review Process File · EMBO Molecular Medicine]

Spirolactone is an Antagonist of NRG1-ERBB4 Signaling and Schizophrenia-Relevant Endophenotypes in Mice

Michael C. Wehr, Wilko Hinrichs, Magdalena M. Brzózka, Tilmann Unterbarnscheidt, Alexander Herholt, Jan P. Wintgens, Sergi Papiol, M. Clara Soto-Bernardini, Mykola Kravchenko, Mingyue Zhang, Klaus-Armin Nave, Sven P. Wichert, Peter Falkai, Weiqi Zhang, Markus H. Schwab and Moritz J. Rossner

*Corresponding authors: Michael Wehr and Moritz Rossner
Ludwig Maximilian University of Munich*

Review timeline:

Submission date:	13 February 2017
Editorial Decision:	29 March 2017
Revision received:	29 May 2017
Editorial Decision:	20 June 2017
Revision received:	26 June 2017
Accepted:	28 June 2017

Transaction Report:

Editor: Céline Carret

1st Editorial Decision

29 March 2017

Thank you for the submission of your manuscript to EMBO Molecular Medicine. We have now heard back from the two referees whom we asked to evaluate your manuscript.

You will see from the comments pasted below, that while referee 2 is more succinct than referee 1, they both agree that the study is interesting and should be published. Referee 1 suggests additional experiments to increase the conclusiveness, and both referees would like to see more details and explanations. Please note that EMBO Molecular Medicine encourages a single round of revision and that, as acceptance or rejection of the manuscript will depend on another round of review, your responses should be as complete as possible.

I look forward to receiving your revised manuscript.

***** Reviewer's comments *****

Referee #1 (Comments on Novelty/Model System):

The approaches are adequate and range from the use of cell culture based HTS screening to in vivo approaches using models to test schizophrenia-like behavior

Referee #1 (Remarks):

In this study Wehr et al develop a cell based HTS-assay to screen for modulators of NRG1-ERBB4 signaling. This is based on the current view that NRG1-ERBB4 hyperactivity contributes to schizophrenia. The authors specifically test approved drugs that offer the chance for drug repurposing. By this approach they identify spironolactone and provide evidence that it acts as a pan-ERBB antagonist that prevents dimer formation. Most importantly, treating NRG-1 TG mice daily with spironolactone ameliorated schizophrenia-like phenotypes.

In sum, this is a very interesting study that summarizes a lot of work ranging from drug discovery to eventually testing the identified drug in a relevant animal model. Taken into account that spironolactone is an approved drug that has however not been considered in the context of schizophrenia, the presented findings are novel and could have immediate impact in the clinic.

I believe this study will be very interesting to the readers of EMBO Mol. Med.

I only have a few questions remaining:

Major issues:

1. Fig 4. It would be helpful to provide quantification for all investigated proteins in addition to ErbB4 and LimK. Also, please provide the number of animals used for the molecular analysis. This is missing for all immunoblot data and would be important since images are not always high quality.

2. Although the authors convincingly show that spironolactone affects ERBB function in cell culture, many different modes of actions have been assigned to it. Foremost spironolactone is known as MR antagonist and at present its questionable if the therapeutic effect in the NRG-1 TG mice is indeed mediated via the NRG1-ERBB pathway (especially since glucocorticoid signaling has been linked to NRG-1 function) . I suggest two approaches to address this issue.

One possibility would be to measure p-ErbB4 as well as p-LimK etc. levels in animals that were used in the experiments described in Fig5 and EV5. Maybe this is what the authors did but at present it is unclear what treatment paradigm was used to generate the data shown in Fig 4C. Moreover, was really the entire mouse brain used for analysis? The description is misleading and sometimes refers to the prefrontal cortex which would of course make more sense.

In addition it would be nice to learn more - maybe in the discussion - about the downstream mechanisms of elevated NRG1-ERBB4 signaling in the context of schizophrenia-like phenotypes. In fact considering that NRG1 is overexpressed from early developmental stages and there is evidence that altered NRG-1 function affect brain development and thereby contributes to the pathogenesis of schizophrenia, it is quite remarkable that even 3 weeks of spironolactone treatment in juvenile/adult mice reverse the disease phenotypes. Do the authors think that this correlates with structural changes?

Along this line it is said that age-matched mice were used for behavioral testing (nothing is said about the mice used for immunoblot analysis). However the age appears to vary from 8-16 weeks. The authors should exclude the possibility that the observed effects are linked to the different ages. Can we exclude the possibility that the effect of spironolactone is mainly linked to the treatment of 8 weeks old mice in which post-natal brain development may not have been completed yet? Is it less efficient in 16 month old mice?

Minor issues:

1. page 3: "...principally offers a fast track to the clinic and has been demanded for SZ (Insel, 2012; Lencz & Malhotra, 2015), and also because many pharma companies withdrew from research..."
the "and" appears misplaced.
2. Page 3: "Instead of changes in protein-coding regions..." should be removed.
3. Fig. 1A: Please indicate the type of cells that were used.
4. At what time of the day was spironolactone injected?

Referee #2 (Comments on Novelty/Model System):

Extremely well executed project

Referee #2 (Remarks):

This is a very interesting study highly deserving of being published especially after minor revision. My only comments re: 1-Authors should explain the differences in concentrations of spironolactone (10 micromolar in electrophysiology vs. 1 in other experiments), 2-explain any sex differences in response to spironolactone

1st Revision - authors' response

29 May 2017

We thank the referees for their constructive comments that greatly improved the quality of the manuscript. All the issues raised by the referees have been addressed (see the point-by-point response below), which prompted us to further analyse existing data that are now included in the updated version of the paper. These data are now shown in Fig. 4C.

In addition, we have expanded our discussion and addressed potential downstream mechanisms of elevated NRG1-ERBB4 signaling in the context of schizophrenia-like phenotypes.

Referee #1

Major issues:

1. Fig 4. It would be helpful to provide quantification for all investigated proteins in addition to ErbB4 and LimK. Also, please provide the number of animals used for the molecular analysis. This is missing for all immunoblot data and would be important since images are not always high quality.

As requested for Fig. 4D, we have now quantified all phospho-protein levels vs. total protein levels of the blots shown in Fig. 4D (was Fig. 4C) using densitometric analysis in ImageJ. As evident from the quantification, phospho-Erk1/2 and phospho-Akt levels were neither changed in *Nrg1*-tg mice, nor in spironolactone-treated animals (for both controls and *Nrg1*-tg mice). In addition, we have added information on the number of independent cellular lysates (n=4, Fig. 4A, B) and lysates from animals (n=2, Fig. 4D, E) used for quantification.

Amendments in the figure legend for Fig. 4:

(B) Quantification of band intensities for phospho-ERBB4 levels (n=4 per condition) shown in (A) using ImageJ. Phosphorylation levels are normalized to protein levels of ERBB4. Data shown as mean, error bars represent SD; t-test, with *, p<0.05, and **, p<0.01.

(C) Experimental design for western blot analysis shown in (D) and (E). *Nrg1*-tg and wt animals were treated daily with spironolactone (50 mg/kg, s.c.) or vehicle (n=2 per genotype and per treatment) for 21 days.

(D) Spironolactone reduces phospho-ErbB4 levels in *Nrg1*-tg mice. Mice were treated with spironolactone for 21 days and sacrificed for western blot analysis. Lysates were probed with indicated antibodies.

(E) Quantification of band intensities for phospho-ErbB4 and phospho-Limk1 levels shown in (D) using ImageJ. Phosphorylation levels are normalized to protein levels of ErbB4 and Limk1. Data shown as mean, error bars represent SD; t-test, with *, $p \leq 0.05$.

2. Although the authors convincingly show that spironolactone affects ERBB function in cell culture, many different modes of actions have been assigned to it. Foremost spironolactone is known as MR antagonist and at present its questionable if the therapeutic effect in the NRG-1 TG mice is indeed mediated via the NRG1-ERBB pathway (especially since glucocorticoid signaling has been linked to NRG-1 function). I suggest two approaches to address this issue.

One possibility would be to measure p-ErbB4 as well as p-LimK etc. levels in animals that were used in the experiments described in Fig5 and EV5. Maybe this is what the authors did but at present it is unclear what treatment paradigm was used to generate the data shown in Fig 4C.

We thank the referee for this insightful comment and her/his suggestions to clearly present spironolactone-mediated changes of cellular signaling observed in vivo.

Indeed, as the referee already assumed, we have addressed p-ErbB4, p-Erk1/2, p-Akt, and p-Limk levels in spironolactone-treated animals and controls (Fig. 4D, was Fig. 4C). As described in the manuscript, major ERBB4 downstream signaling activities that include MAPK signaling (monitored by p-Erk1/2) and PI3K/Akt signaling (monitored by p-Akt) did not show differences in *Nrg1*-tg mice as well as spironolactone-treated mice. However, we have, as stated, noticed differences in p-Limk levels in spironolactone-treated mice, a finding that may indeed be linked to cytoskeleton dynamics and the regulation of synaptic spine morphology and function (Meng et al, 2002, 2004) (see below).

The treatment paradigm for the western blots shown in Fig. 4D was chosen alike the paradigm for the behavioral experiments shown in Fig.5 and EV5. To clarify the experimental situation, we added Fig. 4C as experimental outline for Fig. 4D.

Animals for the behavioral and biochemical analysis were, however, cohorted separately. For the biochemical analysis, animals were sacrificed after 21 days of daily spironolactone treatment, which coincided with the start of the behavioral tests. Mice used for biochemistry were also age-matched (11-13 weeks, with males used only), and the prefrontal cortex region was isolated for sample preparation as stated now in the manuscript.

To clarify the description, we have added a paragraph to the section “Supplementary Material and Methods”, subheading “Protein lysates and western blotting” in the Appendix and the main text. Amendments to the Appendix, section Supplementary Material and Methods, subheading “Protein lysates and western blotting”:

For the biochemical analysis of spironolactone-treated mice (for a precise description of the injection paradigm, see Materials and Methods section, subheading “Mouse behavior analysis”, “Spironolactone treatment”), vehicle control or spironolactone were subcutaneously injected daily for 21 days into age-matched (11-13 weeks) male mice prior to preparation of the mouse prefrontal cortex (n=2 per genotype and treatment). For the generation of lysates, the isolated tissue was immediately placed into cooled (4°C) sucrose buffer (320mM sucrose, 10 mM Tris-HCl, 1 mM NaHCO₃, 1 mM MgCl₂, supplemented with 10 mM NaF, 1 mM Na₂VO₄, 1 mM ZnCl₂, 4.5 mM Na₄P₂O₇ as phosphatase inhibitors, and the complete protease inhibitor cocktail (Roche)), homogenized using an ultra-turrax (IKA GmbH, Staufen, Germany), sonicated (3 pulses for 10 sec) and denatured for 10 min at 70°C in LDS sample buffer.

Amendments to the main text:

We tested whether phospho-ERBB4 levels were also regulated by chronic spironolactone treatment in *Nrg1*-tg mice and injected the drug for 21 consecutive days before sacrificing the mice for biochemical analysis (Fig 4C).

In lysates from mouse prefrontal cortex, phospho-ErbB4 levels were efficiently visualized using the p-ERBB4-Y1284 antibody (Figs 4D and E).

Moreover, was really the entire mouse brain used for analysis? The description is misleading and sometimes refers to the prefrontal cortex which would of course make more sense.

We have corrected the description of the tissue sampling, which indeed was always from frontal regions (see paragraph above)

In addition it would be nice to learn more - maybe in the discussion - about the downstream mechanisms of elevated NRG1-ERBB4 signaling in the context of schizophrenia-like phenotypes. In fact considering that NRG1 is overexpressed from early developmental stages and there is evidence that altered NRG-1 function affect brain development and thereby contributes to the pathogenesis of schizophrenia, it is quite remarkable that even 3 weeks of spironolactone treatment in juvenile/adult mice reverse the disease phenotypes. Do the authors think that this correlates with structural changes?

We like to refer the reviewers comment also to the following paragraph in the Discussion which describes the potential link between *Nrg1* function, and a potential impact of spironolactone treatment on the structural changes:

‘Our biochemical analysis implicates LIMK1 signaling, but not ERK1/2 nor AKT1 as potential downstream effectors of spironolactone treatment in *Nrg1*-tg mice. As a non-receptor protein serine/threonine kinase, LIMK regulates synaptic spine morphology and function by modulating cytoskeleton dynamics (Meng *et al*, 2002, 2004; Bennett, 2011). Further, LIMK1 has been linked to NRG1 signaling and SZ-relevant endophenotypes in a *Nrg1*-tg mouse model (Yin *et al*, 2013). We show that phospho-LIMK1 levels were upregulated in *Nrg1*-tg mice treated with spironolactone suggesting that LIMK1 activity may possibly integrate spironolactone’s inhibitory effect by promoting spine enlargement, and thus synapse formation, through controlling actin cytoskeleton dynamics.’

According to the referee’s suggestion, we have substantiated the discussion on structural changes by adding the following sentences to the corresponding paragraph:

Nrg1-tg animals display subtle structural changes related to spine morphology, i.e. the number of bifurcated spines is increased (Agarwal *et al*, 2014). Therefore, it might be possible that spironolactone treatment reverts this structural endophenotype. Nonetheless, the increased levels of p-LIMK1 rather favors a mechanism compensating for the structural changes in *Nrg1*-tg mice, which may underlie network disturbances in these animals, by stimulating structural plasticity via increased LIMK1 activity. To further explore the mode-of-action of spironolactone in the future, it’s impact on structural plasticity should be addressed in additional studies.

We like to state that these highly interesting follow-up experiments would best be performed with high-resolution imaging of living mice using e.g. a STED setup and is beyond the scope of this manuscript.

Along this line it is said that age-matched mice were used for behavioral testing (nothing is said about the mice used for immunoblot analysis). However the age appears to vary from 8-16 weeks. The authors should exclude the possibility that the observed effects are linked to the different ages. Can we exclude the possibility that the effect of spironolactone is mainly linked to the treatment of 8 weeks old mice in which post-natal brain development may not have been completed yet? Is it less efficient in 16 month old mice?

The referee points out that *Nrg1* is a critical regulator of brain development and that *Nrg1* overexpression may have different effects on behavior, e.g. in juvenile vs. adult mice. We would like to notice that we consider mice between 8 and 16 weeks as adult mice. Nonetheless, to address any age-related effects on the behavioral effects observed, both for spironolactone-treated and control-treated animals, we grouped mice into juvenile (8-11 weeks) and adult (12-16 weeks) categories. A covariate analysis using ANCOVA was performed for the open field test, Y-maze test and light-dark preference test, which indicated that age had no significant effect on the genotype-dependent treatment response (new Fig S3 in the Appendix). We also stated this in the main manuscript, by adding the following sentence to the results section describing the behavioral results: As *Nrg1* is a critical regulator for brain development, we aimed to exclude age-related consequences on the behavioral effects observed using a covariate analysis, which links age with test performance. To do this, we grouped mice into juvenile (8-11 weeks) and adult (12-16 weeks) categories, and

ANCOVA with the covariate age did not reveal an effect on the genotype-dependent treatment response (Appendix, Fig S3).

In addition, we include below a figure displaying all pairwise t-test p-values between juvenile and adult *Nrg1*-tg mice, which are not significant within each experimental condition for comparisons (as stated in the Appendix Fig S3 descriptions), which further supports the absence of an age-dependent effect. The layout of this figure R1 is corresponding to Fig S3 in the Appendix, with all T-test values depicted.

Minor issues:

1. page 3: "...principally offers a fast track to the clinic and has been demanded for SZ (Insel, 2012; Lencz & Malhotra, 2015), and also because many pharma companies withdrew from research..." the "and" appears misplaced.

Corrected.

2. Page 3: "Instead of changes in protein-coding regions..." should be removed.

As suggested, the part is removed.

3. Fig. 1A: Please indicate the type of cells that were used.

This illustration relates to the repurpose screening, for which PC12 cells were used in a co-culture setting. We have amended the figure legend accordingly. The first sentence now reads:

(A) The ERBB4-PIK3R1 split TEV assay monitors NRG1-ERBB4 signaling in PC12 cells.

At what time of the day was spironolactone injected?

As described in the Materials and Methods section, subheading "Spironolactone treatment", mice were injected daily in the afternoon. The description was already described before and reads:

'To avoid injection-induced stress prior to behavioral testing, mice were injected in the afternoon, after the entire cohort has completed the behavioral paradigm.'

Referee #2 (Remarks):

1-Authors should explain the differences in concentrations of spironolactone (10 micromolar in electrophysiology vs. 1 in other experiments)

For both electrophysiology and biochemistry experiments, we applied spironolactone at a concentration of 10 μ M. To address the effect caused by spironolactone in the presence of canrenone, we have lowered the concentration of spironolactone to 5 μ M (Fig. 4H). The higher concentrations of spironolactone (5 and 10 μ M) in the electrophysiological experiments were chosen to be above the IC50 of about 1 μ M as determined by the reporter gene assays to avoid reduced efficacy by higher absorbance and less penetrance of slices. Moreover, the application in the electrophysiological experiments was acute (in the min range) which, together with the absence of toxic effects in the reporter assays (up to 24h incubation), argues against any side effects of the chosen higher concentrations, e.g. by an impact on viability.

2-explain any sex differences in response to spironolactone

In our analyses, both for behavior and biochemistry, we have only used male mice and thus cannot comment on potential sex effects. We kindly ask the referee to refer to the section “Materials and Methods”, subheading “Mouse behavior analysis”. The description was already described before and reads:

‘For behavioral testing, age-matched male mice (8-16 weeks) on C57Bl/6 background that constitutively overexpress the 2 \times HA-tagged Nrg1 type III b1a isoform (Nrg1-tg) under the control of the mouse Thy1.2 promoter (Velanac et al, 2012) and their wild type (wt) littermates as controls were used.’

2nd Editorial Decision

20 June 2017

Thank you for the submission of your revised manuscript to EMBO Molecular Medicine. We have now received the enclosed reports from the referee who was asked to re-assess it. As you will see the reviewer is now globally supportive and I am pleased to inform you that we will be able to accept your manuscript pending following final editorial amendments.

***** Reviewer's comments *****

Referee #1 (Comments on Novelty/Model System):

As mentioned in the original review the manuscript present a very interesting and innovative approach to screen for drugs that might help to treat schizophrenia. Via the identification of spironolactone the authors show prove of concept. Using mice for this type of research is adequate.

Referee #1 (Remarks):

I have not further remarks. The authors have addressed all of my previous concerns sufficiently.

2nd Revision - authors' response

26 June 2017

Authors made the requested editorial changes.

Corresponding Author Name: Michael Wehr, Moritz Rossner

Manuscript Number: EMM-2017-07691